# The Growth–Climate Relationships of Three Dominant Subalpine Conifers on the Baima Snow Mountain in the Southeastern Tibetan Plateau

**DOI:** 10.3390/plants13121645

**Published:** 2024-06-14

**Authors:** Siyu Xie, Yun Zhang, Yaoyao Kang, Tao Yan, Haitao Yue

**Affiliations:** 1College of Ecology and Environment, Southwest Forestry University, Kunming 650224, Chinazhangyuncool@163.com (Y.Z.);; 2Beijing Forestry and Parks Planning and Resource Monitoring Center, Beijing 101118, China

**Keywords:** tree rings, climate change, basal area increment, upper distributional limit, high mountain forest

## Abstract

The impact of climates on the radial growth of muti-species remains insufficiently understood in the climate-sensitive southeastern Tibetan Plateau, and this hampers an effective assessment of forest growth under the background of global warming. Here, we studied the growth–climate relationships of three major species (*Abies georgei*, *Larix potaninii*, and *Picea likiangensis*) on the Baima Snow Mountain (BSM) by using dendrochronology methods. We constructed basal area increment (BAI) residual chronologies based on the dated ring-width measurements and correlated the chronologies with four climate factors. We also calculated the contributions of each climate factor to species growth. We found that photothermal conditions played a more important role than moisture in modulating radial growth, and *P. likiangensi* presented the strongest sensitivity to climate change among the three species. The growing season (June and July) temperature positively affected the radial growth of three species. Winter (previous December and current January) SD negatively impacted the tree growth of *A. georgei* and *P. likiangensis*. Significant correlations between growth and precipitation were detected only in *A. georgei* (January and May). Warming since the beginning of the 1950s promoted the growth of *A. georgei* and *P. likiangensis,* while the same effect on *L. potaninii* growth was found in the recent 50 years.

## 1. Introduction

Forests cover approximately one-third of the Earth’s land surface and regulate global carbon, water, and energy cycles [1]. Terrestrial forest productivity and carbon budget are affected by climate change, such as rising temperatures, extreme drought, and elevated CO_2_. Understanding the relationship between climate drivers and forests is crucial for predicting the potential impacts of future climate change on terrestrial ecosystems.

Complex mechanisms exist in the tree growth response to climate change. The start of the growing season of high-elevation forests has advanced under global warming and caused the enhancement of tree growth and forest productivity [2]. However, warming-induced drought widely affected tree physiological processes and limited forest growth, and even determined the survival of trees in water-deficit regions [3,4,5]. Beyond that, the spatial patterns of tree growth responses to climate change varied on broad spatial scales [6,7]. At finer regional scales, tree growth sensitivity to climate was corrected by habitat-specific (i.e., variations in topographic position and soil properties) and species-specific. Species may respond differently to climate conditions, even when growing together on a common site [8,9,10]. For instance, in hilly and mountainous areas, elevation affects the microclimate, such as the temperature, humidity, and soil moisture, which in turn influences the relationship between the climate and tree growth [11]. Villalba and Veblen [12] reported that the growth responses of different tree species to climate change were distinct, even in the adjacent subalpine zone of the Colorado Front Range.

To better predict growth–climate relationships in forest ecosystems, the multi-species and site-specific growth responses to climate change need to be considered [13,14], particularly in climate-sensitive areas. Subalpine forests growing in specialized alpine ecosystems, with fir and spruce as the dominant species, are particularly sensitive to climate change. Such forests play an extremely important and irreplaceable role in the conservation of fragile ecosystems, biodiversity, water resources, and carbon sequestration [15]. Previous studies have suggested that tree growth responses to climate change in the subalpine area were affected by both moisture [16,17,18] and temperature [19,20,21], depending on the local limitation factor and tree species. 

The alpine and subalpine regions of southwest China primarily refer to the high-altitude areas, including the western Sichuan, northwestern Yunnan, and southeastern Tibetan Plateau, where forest ecosystems are dominated by subalpine coniferous forests at elevations ranging from 2500 m a.s.l. to 4500 m a.s.l. [22]. The region often features a complex topography and variable climatic conditions characterized by strong gradients in temperature and rainfall as well as their interactions at relatively short distances [23]. The subalpine forest in the area is dominated by many conifer species, and the area has been experiencing a warming trend [24], which provides an ideal place for carrying out dendrochronology studies. 

The Baima Snow Mountain (BSM) is one of the typical snow mountains in the southeastern Tibetan Plateau and has a large area of primary subalpine forests, which provide unique opportunities to study the growth–climate relationships of multi-species. However, previous dendroclimatology studies in this region remain scarce and inconsistent [25,26], reflecting the complexity of species-dependent growth responses to climate, due to their different physiological adaptations and growth strategies. This calls for a more thorough understanding of the relationships between the tree growth of multiple species and climate variables [27].

Dendrochronology methods were utilized to assess the growth sensitivity of three subalpine coniferous species (*Abies georgei*, *Larix potaninii*, and *Picea likiangensis*) to climatic variability on the BSM. This study aimed to identify the major climate drivers influencing tree radial growth and quantify their relative importance among the investigated species in the study area. We hypothesized that (1) the radial growth of studied species in the growing season would benefit from warming, whereas negative impacts of temperature on growth were expected in dry periods, and that (2) temperature and precipitation interacted temporally, with tree growth during the early growing season being primarily limited by precipitation and soil availability, and being constrained by temperature in the late growing season.

## 2. Materials and Methods

### 2.1. Study Area

The study area (27°25′–28°36′ N, 98°47′–99°21′ E) is located in the Baima Snow Mountain National Nature Reserve of Deqing, Shangri La Tibetan Autonomous Prefecture, southwest China (Figure 1), which has experienced a warming trend over the recent few decades (Appendix A). The climate is controlled by the southwest monsoon, with characteristics of simultaneous rain and heat, and distinct dry and wet seasons. The annual average temperature and annual total precipitation are around 6.4 °C and 746 mm, respectively. Generally, 63% of the annual precipitation is concentrated between June and September; annual average sunshine duration is 1956.7 h. July temperature is the warmest (13.2° C), while January temperature is the coldest (−1.3° C) (Figure 2). The main vegetation types on the BSM include mixed coniferous and broad-leaved forests (3000–3200 m), subalpine dark coniferous forests (3200–4000 m), and alpine scrubs and meadows (above 4000 m).

### 2.2. Climate Data

Monthly mean temperature and precipitation data (1901–2022) were collected from the grid nearest to our sampling sites, which were obtained from the Climatic Research Unit at a resolution of 0.5° × 0.5° [28]. Daily sunshine duration (SD) data (1954–2020) were downloaded from Deqin stations near our sample sites from the National Meteorological Information Centre (NMIC) of China [29]. Daily soil moisture content (SM) (1948–2014) in the root zone (0–100 cm) was collected from the NASA Global Land Data Assimilation System at a resolution of 0.25° × 0.25° [30]. We calculated the values of seasonal climate variables by summing monthly total precipitation and averaging monthly mean temperature, monthly total SD, and monthly total SM for the early growing season (from March to May), growing season (from June to August), and late growing season (from September to October).

### 2.3. Tree-Ring Sampling and Processing

We sampled mature and healthy trees without insect damage at natural forests (Table 1). One or two cores per tree from the opposite direction were extracted at a height of 1.3 m with a 5.15 mm diameter increment borer. Obtained cores were labeled and stored in plastic straws to prevent breakage. In the laboratory, the cores were glued to the grooved wooden bars by using white latex and tape to prevent bending during air drying. Transverse surfaces of the samples were polished with sandpaper successively fine grits (120 mesh to 800 mesh) until the ring boundaries were clearly visible. Then, the cores were visually cross-dated under a microscope and scanned by EPSON (Expression11000XL, Seiko Epson Corporation, Suwa, Japan) scanner. The scanner’s parameters were set to 24-bit full-color professional mode image types, with a resolution of 2000 dpi. The tree-ring widths were measured using CDendro and CooRecorder ver. 9.8.1 software at a precision of 0.001 mm. Furthermore, the results were verified with COFECHA [31] to strengthen the cross-dating between trees. Cores with inadequate quality were removed based on the testing results. Ultimately, 88 trees from 160 cores remained for further analysis.

### 2.4. Chronology Development

The tree-ring width series were converted into basal area increments (BAI), which is more representative of overall tree growth than ring widths [32,33]. The structure of BAI is as follows:(1)$$BAI_{t}=\pi R_{t}^{2}-\pi R_{t-1}^{2}$$
where *t* is the year of tree-ring formation and R is the radius of the tree in year *t*. The calculation used the function ‘bai.out’ in R package ‘dplR’ [34]. To eliminate intrinsic growth trends related to age and size in BAI and compare the annual growth variability of species, BAI was detrended using generalized additive mixed models (GAMMs) [35]. One model was constructed for each species (see Appendix A about the BAI residuals chronologies and diagnostic plots of the GAMMs). A logarithmic transformation was applied to improve the normality of BAI value distributions. The structure of GAMM is as follows:(2)$$log({BAI}_{ijt})=s(log({BA}_{ijt}))+s({Age}_{ijt})+{(CoreID}_{ij})+cor{AR1}_{ij}$$
where *i* represents the individual core, *j* represents the species, and *t* represents the year. BA is the basal area of core *i* at specific year *t*, computed as the sum of BAI of previous years. Age represents cambial age, an estimation of tree age based on the ring counts, and *s* refers to the smooth terms of the GAMM in which the cubic regression spline is used. BA and Age are fixed effects, and CoreID is a random effect. Considering the time-series characteristics of our measurement data, an AR1 (*p* = 1, *q* = 0) correlation structure was incorporated to account for temporal autocorrelation with the CoreID as the grouping factor. The model was fitted using the ‘mgcv’ package in R [36]. We calculated BAI residuals to represent the tree growth index [37]:(3)$${BAI}_{residual}=\log\left({BAI}_{observed}\right)-log({BAI}_{predicted})$$

### 2.5. Growth–Climate Relationships

The response function was applied to analyze growth–climate relationships using DendroClim2002 [38]. The correlation coefficient between the BAI residual chronologies and climate variables was calculated with the common interval (1935–2019) of all chronologies and available time series of meteorological data. To capture the lag effect of climate on radial growth [39], 16 months of meteorological data spanning from the previous year’s July through the following October were used. We selected four climate variables, including monthly mean temperature, total precipitation, SD, and SM. A moving interval analysis between BAI residual chronologies and monthly mean temperature as well as total precipitation was conducted with 35-year windows to investigate the temporal stability and consistency of the growth–climate relationships by using DendroClim2002 [38].

Moreover, for each chronology, we constructed a multiple regression model to evaluate additive effects on BAI of interannual variability in seasonal temperature, precipitation, SD and SM (summed climate variables). Then, we conducted variance inflation factors (VIF) on all models to check for collinearity among climate variables and found values to be lower than 2.6 (VIF < 4). To compare the contribution of each seasonal climate variable to the tree growth in the early and late growing seasons, respectively, we assessed the relative importance of the regressors using the ‘relaimpo’ package in R [40]. It considers the dependence of regressors and has an advantage over the use of *R*^2^ from univariate regressions [40].

## 3. Results

### 3.1. Climate Drivers at Monthly and Seasonal Scale

The BAI residual chronologies from three species demonstrated commonality and species specificity with monthly climate variables (Figure 3). *L. potaninii* growth was affected only by temperature, showing significant and positive correlations with the current June temperature (*p* < 0.05). For *P. likiangensis*, temperature and SD drove its radial growth. The current September temperature positively influenced its radial growth, while SD had negative impacts on *P. likiangensis* growth in previous December and current January but significant and positive effects in current September (*p* < 0.05). Regarding *A. georgei*, three climate factors affected its radial growth. The current July temperature positively influenced its growth (*p* < 0.05). *A. georgei* growth was also positively correlated with precipitation in January and May, while it showed a significantly negative response to SD in previous August and January (*p* < 0.05). Compared with the impact of temperature, precipitation, and SD, three species appeared to have much weaker relations to SM, without showing any significant correlations.

In the early growing season, both *A. georgei* and *P. likiangensis* responded positively to precipitation and SM, and the positive effects were maintained until the growing season (Figure 4). While *L. potaninii* showed consistently positive correlations with precipitation and SM in the late growing season, it showed negative associations with temperature and SD during the period. Furthermore, the promotion of *A. georgei* growth by precipitation and SM in the early growing season shifted to inhibition in the late growing season, which was also found in the growth responses of *P. likiangensis*. The growth responses of *P. likiangensis* to temperature shifted from negative to positive from the early to late growing season.

### 3.2. Relative Importance of Climate Variables

An assessment of relative importance revealed that the main explanatory variables of BAI residual chronologies differed between seasons (Figure 5). The contribution of temperature to the radial growth of three species was highest during the growing season, reaching a maximum of 73%. SD contributed most to *L. potaninii* growth (30%), followed by precipitation (27%) during the early growing season, while precipitation (47%) played a more significant role in the late growing season. For *A. georgei*, SD (57%) was also the most influential factor during the early growing season, followed by temperature (25%), which became more important (47%) than SD (30%) in the late growing season. *P. likiangensis* showed a strong reliance on temperature (49% and 67%) during both the early and late growing seasons, with other factors contributing less to growth. SM played a minor role in regulating the growth of species in our study sites, contributing the least to the model, except for *L. potaninii* in the early growing season and growing season.

### 3.3. Growth–Climate Relationships at Inter-Annual Scale

The results of temporal stability analysis were in accordance with response function analysis, with *P. likiangensis* presenting more stable correlations with climate factors among the three species. For temperature, *A. georgei* showed stable relationships with July temperature during the period 1952–1995 by exhibiting significant and positive correlations (Figure 6A). The temperature rise in June in the recent 50 years (since 1973) promoted the radial growth of *L. potaninii* (Figure 6B). For *P. likiangensis* (Figure 6C), the positive correlation with current September temperature has been stable since 1952. Although significant correlations with previous September and current August appeared in some episodes, this positive response was not detected in the general response (Figure 3). 

Unlike the temperature, the temporal stability of growth responses to precipitation was weaker, showing fewer significant correlations (Figure 7). The positive effect of March precipitation on *A. georgei* growth was strong between 1943 and 1986, while January and May precipitation significantly affected its growth in some decades (Figure 7A). *L. potaninii* was inhibited by April precipitation from 1943 to 1987 (Figure 7B) and did not exhibit any general responses. *P. likiangensis* showed a significant response to previous September and current March temperatures in some periods between 1938 and 1984 (Figure 7C). 

## 4. Discussion

Three species showed similarities and differences in their responses to climate change, probably due to their biological characteristics. *A. georgei* and *P. likiangensis* are characterized as shade-tolerant and moisture-loving, while *L. potaninii* is typically considered to be light- and heat-loving. Therefore, the similarity between *A. georgei* and *P. likiangensis* in their responses to climate change is higher. In addition, since sampling sites were near their upper distributional limit, photothermal conditions contributed more to radial growth than precipitation. We discussed important findings of common and species-specific responses to the climates outlined below.

### 4.1. Common Response

Monthly temperature positively affected the radial growth of *L. potaninii* (June), *A. georgei* (July) and *P. likiangensis* (July) in the growing season (Figure 3), indicating that a higher temperature during the growing season may stimulate the radial growth of these three species. This was further confirmed by the results of the climatic effects on tree growth in the growing season and the relative importance of each climatic factor, as evidenced by the positive correlations that were found with temperature for *L. potaninii* and *P. likiangensis* growth (Figure 4), with the highest contribution of temperature to tree growth observed during the growing season (Figure 5). Summer temperature provided sufficient warmth to favor the cell production rate and accumulate photosynthates, inducing wider tree rings [20,41]. The production of non-structural carbohydrates (NSCs), required for radial growth in trees, relies on foliar photosynthesis, which depends on the function of chlorophyll and photosynthetic enzymes. An increase in temperature during the growing season can accelerate the function of photosynthetic enzymes, potentially enhancing the storage of carbohydrates needed for xylem growth [42]. The contribution of summer temperature to tree growth was commonly found in high-altitude stands in the Hengduan Mountains [43,44]. 

SD was also decisive in controlling tree growth by influencing spring phenology (budburst and leaf unfolding) and autumn phenology (growth cessation and bud set preceding dormancy) [45]. An early winter SD had negative impacts on *A. georgei* (January) and *P. likiangensis* (previous December and current January) growth (Figure 3). The high SD in the two months may increase winter desiccation and delay stem rehydration or even cause winter injuries, which could have a detrimental impact on cambial activity, resulting in growth beginning later than usual [46]. This phenomenon can be explained by two possible reasons. First, a greater SD in winter was unconducive to the development of permafrost and snow cover, a main source of soil water for radial growth in the upcoming growing season [47]. The increase in SD was accompanied by a rise in the duration of solar radiation, thereby providing more heat to speed up the snow-melting process. Second, a high SD can induce elevated surface temperatures, especially in plateau regions. Warming during winter can consume the nutrients stored in the early stages and reduce the availability of nutrients for the growing season [48]. However, significant effects of SD on *L. potaninii* growth were not detected. Thus, the species is shade-intolerant and is not sensitive to variations in SD. 

According to the correlation analysis between seasonal climate factors and tree growth (Figure 4), *L. potaninii* responded differently compared to the other two species. Humid conditions were more favorable for *A. georgei* and *P. likiangensis* growth (i.e., positive correlation with SM and precipitation) in the early growing season and growing season, with warm conditions (i.e., positive correlation with SD and temperature) were favorable in the late growing season. The climate response of *L. potaninii* was exactly the opposite. This discrepancy may indicate that the differences in tree species lead to different mechanisms of growth responses to climate change. Conducting more detailed physiological experiments or considering non-climate factors can further help reveal its mechanism. Particularly for *L. potaninii*, its radial growth was also affected by Larch budmoth (*Zeiraphera diniana*) on the BSM, which could regulate tree growth at a monthly scale [49]. 

### 4.2. Species-Specific Response

*P. likiangensis* may be able to benefit from a prolonged length of the growing season due to the warmth enabling it to maintain cambial activity and cell differentiation processes as it showed a significant and positive correlation with September temperature and SD (Figure 3). Warm conditions in the late growing season might increase carbohydrate storage in the stem and thus favor wide ring formation [50]. The promoting effect of autumn temperature on tree growth in the current year was consistent with the study of *P. likiangensis* in the Potatso National Park near our sampling sites [51]. Furthermore, SD had a strong interaction with temperature. Temperature allowed the species to maintain metabolic activity during cell production and differentiation, and SD acted as a signal to regulate the timing of the growth rate and synchronize radial growth [52]. Tracheid production was influenced by temperature in a cold environment, with a rise in the SD and the ongoing warm temperature in September potentially leading *P. likiangensis* to continue sustaining high rates of cell division in the late growing season [53]. Previous studies also reported a positive correlation between lignin content in the secondary cell wall of the terminal latewood tracheid of spruce and temperature in September [54].

A significant positive influence of the precipitation in January and May (*p* < 0.05) on the radial growth of *A. georgei* was also observed (Figure 3). The sensitivity of *A. georgei* to winter precipitation was probably related to facilitating the formation of a thicker snow cover, which effectively protected plant roots from the potential damage induced by low temperatures and harsh winds at high altitudes, particularly for shallow-root species like *A. georgei* [55]. The positive growth–precipitation relationships mainly arose from the compensatory effect of May precipitation in the dry season, when the temperature rose rapidly. This may have caused a water deficit for tree growth that was induced by the acceleration of plant transpiration and soil moisture evaporation, which could have limited photosynthesis and reduced the assimilation and reserves of carbohydrates in trees [56]. May precipitation could compensate for the loss of water in the soil caused by rapid increases in temperature (Figure 2), alleviate plant transpiration, and avoid stomatal closure. Moreover, precipitation could supply moisture to the soil for the onset of xylogenesis and xylem cell production during the cambium active period of tree growth [57]. The importance of precipitation in the early growing season has also been observed for several subtropical conifers in the Yunnan-Guizhou Plateau [58].

### 4.3. Dynamics of Growth–Climate Relationships

There was not a high and stable relationship between tree growth and climate factors throughout the whole studied period, revealing that the area lacked a determined climate variable influencing the three species. However, 1952 may be an important year as it showed positive effects on tree growth on the BSM due to higher temperatures, with some significant correlations being established that year (Figure 6). The warming after the 1950s might have triggered interannual fluctuations in growth–climate relationships in our study area (Appendix A). The variation in correlation patterns over time between radial growth and climate variables suggested that the dynamic response relationships were affected by the changing climate, as has been reported for subalpine forests in previous studies [24,57]. Among the three species, the interannual dynamic relationship of *P. likiangensis* was more stable (higher significant correlation values and more significant responses), indicating a higher sensitivity to climate change. 

## 5. Conclusions

In our analysis, temperature interacted with the photoperiod and stood out as an important growth driver for the tree species on the BSM. We found signs of heightened sensitivity to temperature in trees, with an intensification of the influence of elevated temperature on growth over the recent several decades. Our results supported the hypothesis that tree growth in cold regions is mostly temperature-limited [27,57,59]. Meanwhile, species’ differences need to be considered to better comprehend the growth–climate relationships regarding global climate change due to the complexity of plant physiological traits. As global warming is projected to exceed 1.5 °C to 2 °C during the 21st century [60], annual growth variability in subalpine forests in the study area would benefit from rising temperatures, based on the understanding of the growth responses of the three species to climate. However, continuous global warming and insufficient precipitation may impose drought stress and limit the radial growth of the subalpine forest in the future.

## Figures and Tables

**Figure 1 plants-13-01645-f001:**
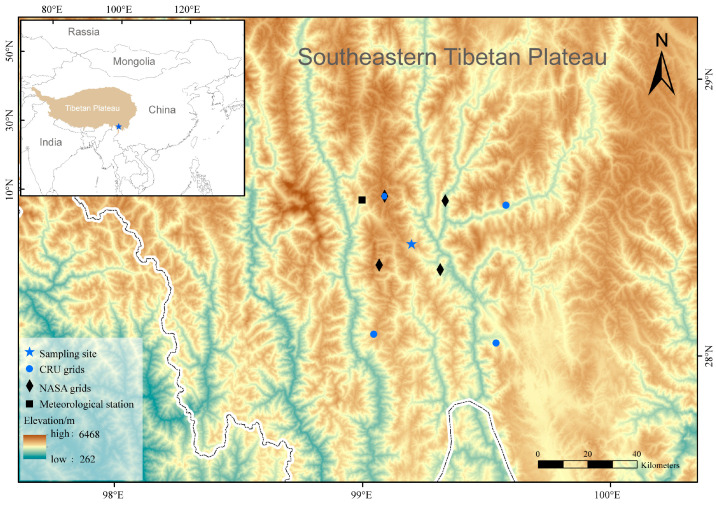
The location of sampling site, CRU grids (temperature and precipitation data), NASA grids (soil moisture data), and meteorological station (sunshine duration data).

**Figure 2 plants-13-01645-f002:**
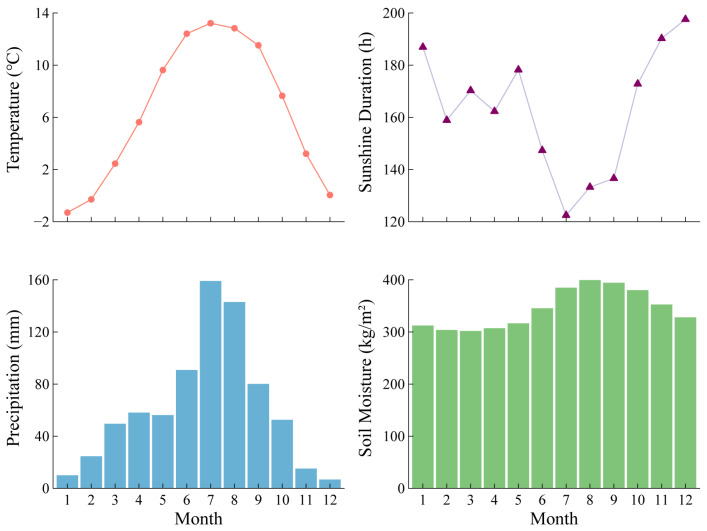
The monthly climatic characteristics at Baima Snow Mountain. Temperature and precipitation (1901–2022); sunshine duration (1954–2020); soil moisture (1948–2014).

**Figure 3 plants-13-01645-f003:**
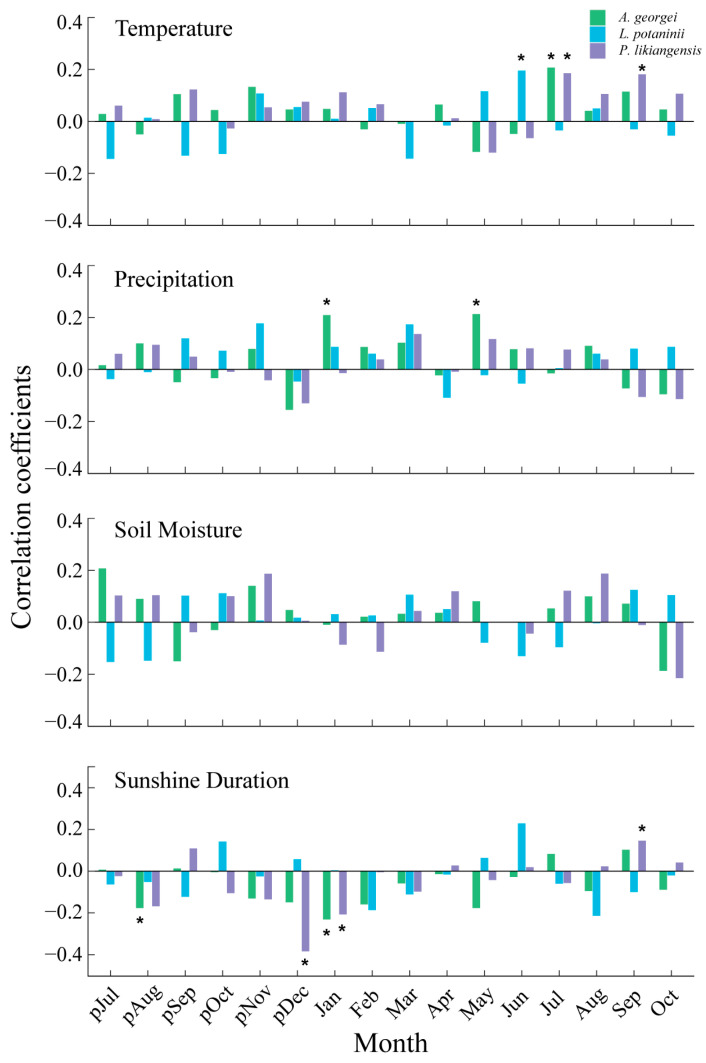
Correlation coefficients between the BAI residual chronologies and temperature (1935–2019), precipitation (1935–2019), sunshine duration (1954–2019), and soil moisture (1948–2014). The same below. * indicates a significant correlation at *p* < 0.05. p: previous year.

**Figure 4 plants-13-01645-f004:**
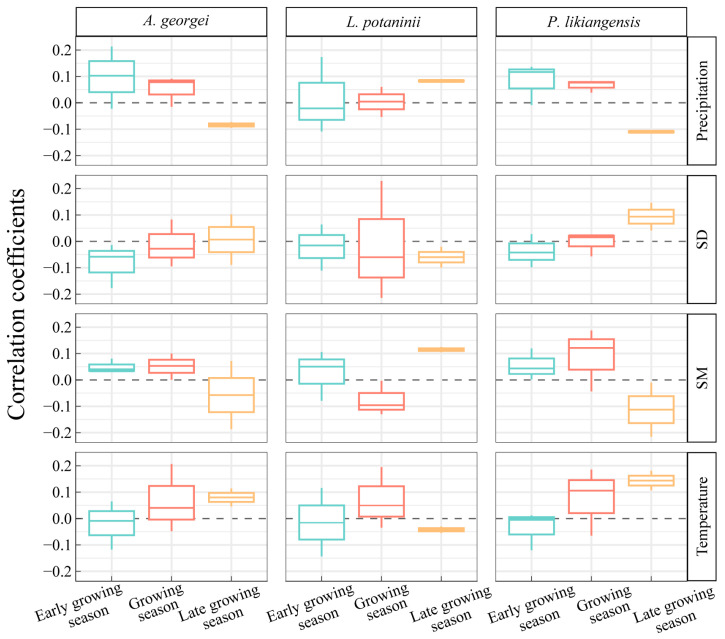
Boxplot of correlation coefficients between BAI residual chronologies and climate variables. Early growing season: March to May; growing season: June to August; late growing season: September to October. The same below.

**Figure 5 plants-13-01645-f005:**
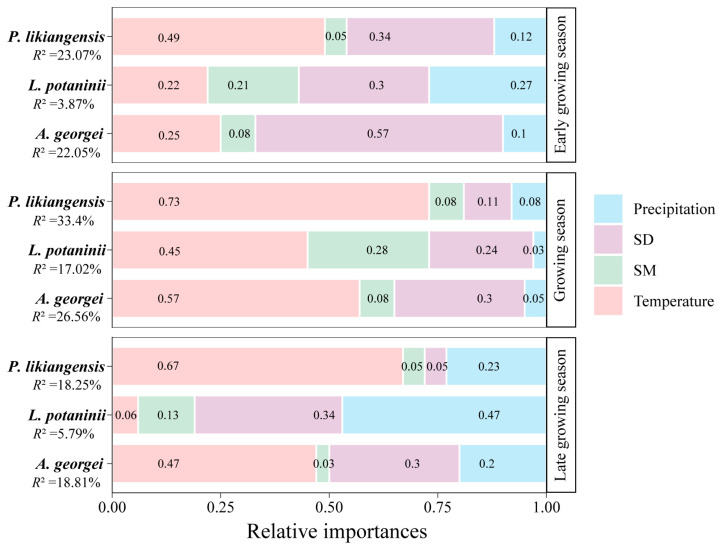
Relative importance of climate variables contributing to BAI per species and per season. The overall explained variance (*R*^2^) for each model is reported underneath each species. Metrics were made to total 100%.

**Figure 6 plants-13-01645-f006:**
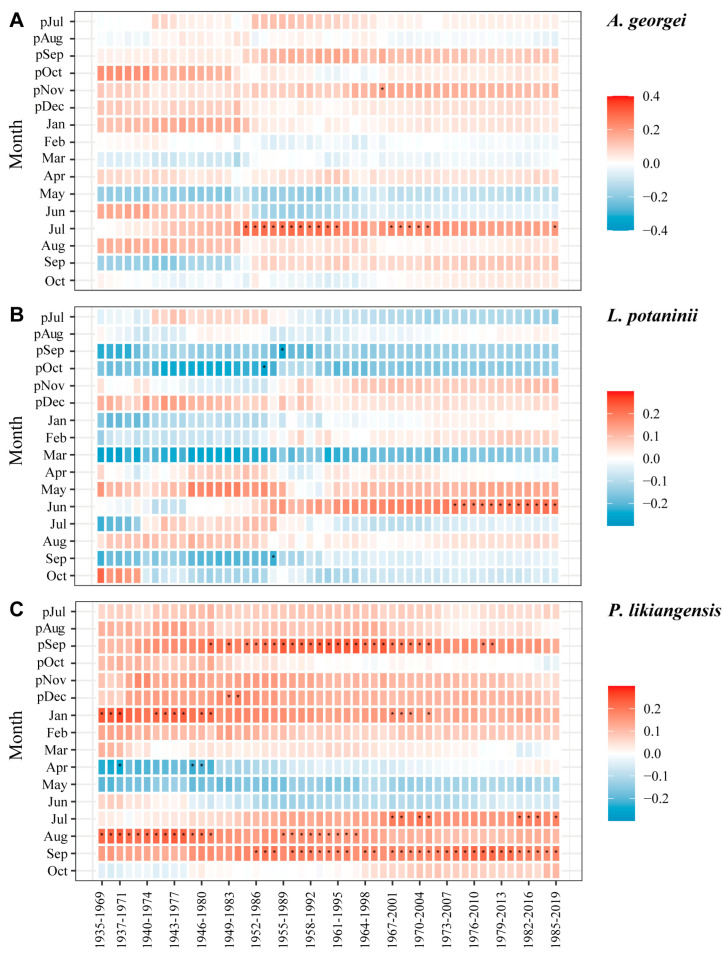
Moving interval analysis between BAI residual chronology of *A. georgei* (**A**), *L. potaninii* (**B**), and *P. likiangensis* (**C**), as well as mean temperature from previous July to current October (1935–2019). * indicates a significant correlation at *p* < 0.05. p: previous year.

**Figure 7 plants-13-01645-f007:**
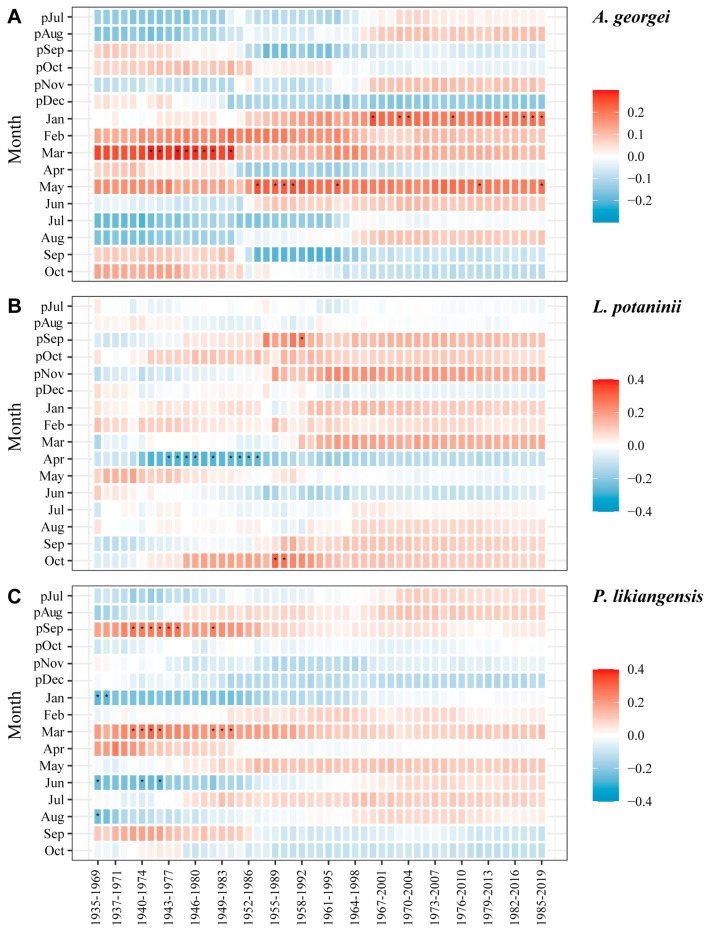
Moving interval analysis between BAI residual chronology of *A. georgei* (**A**), *L. potaninii* (**B**), and *P. likiangensis* (**C**), as well as precipitation from previous July to current October (1935–2019). * indicates a significant correlation at *p* < 0.05. p: previous year.

**Table 1 plants-13-01645-t001:** Sampling information.

Species	Longitude (E)	Latitude (N)	Altitude (m)	No. (Trees/Cores)		Time Span	Common Interval
				Total	Used		
*Abies georgei*	99°7′36″	28°19′39″	4169	39/78	38/68	1869–2019	1935–2019
*Larix potaninii*	99°7′24″	28°19′42″	4141	30/60	29/53	1687–2019	
*Picea likiangensis*	99°7′31″	28°20′58″	3735	22/44	21/39	1698–2019	

## Data Availability

The original contributions presented in the study are included in the article and Appendix A, further inquiries can be directed to the corresponding author.

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
