# Peer review of "The Growth–Climate Relationships of Three Dominant Subalpine Conifers on the Baima Snow Mountain in the Southeastern Tibetan Plateau"

_plants, 2024, doi:10.3390/plants13121645_

Round 1

Reviewer 1 Report

Comments and Suggestions for Authors

The manuscript presents an in-depth analysis of the Growth-Climate Relationships of Three Dominant Subalpine Conifers on Baima Snow Mountain in the southeastern Tibetan Plateau. It delves into the growth-climate relationships of three major species, namely Abies georgei, Larix potaninii, and Picea likiangensis, within the Baima Snow Mountain (BSM) region. The primary objective of this study was to identify the principal climate drivers influencing tree radial growth and to assess their relative significance across the studied species in the area. Additionally, the author proposed two hypotheses, which were thoroughly investigated and validated within the manuscript. The manucrispt is quite detailed, some minor suggestions below:

2.2. Climate data - move the links to references;

I suggest adding formula numbering; And separate all calculation equations into a new line (line 155);

The results section provides a comprehensive description. While I have no significant comments, it is advisable to adjust the figures to ensure that they do not appear crowded to the side and contribute to an overall clear presentation.

Reviewer 2 Report

Comments and Suggestions for Authors

In recent decades, dendrochronology has been performed more and more in China, but each new example is still an important contribution.  This study was done at an unusual site (line 68), and at a very high elevation (Table 1), making it all the more remarkable.

Some comments for minor revision:

Lines 168-169: trees’ breast height.  Sorry, but trees don’t have breast height.  This expression is just a forestry convention for indicating about 1.5 m off the ground.

Line 134: tree-ring measurements from each tree were averaged.  Cook (1990, in the Cook and Kairiusktis book) said that averaging raw ring widths across trees is generally not done in dendrochronology, because raw ring-width series are not stationary through time.  Averaging across trees is usually done after detrending or otherwise standardizing raw ring-widths, perhaps after BAI calculation.

Chronology: do we not get to see time series plots of the tree-ring chronologies?

Fig. 3: Any mechanism for SD of December affecting ring widths the following summer at sites at or above 4000 m in elevation?  Statistical significance is one thing, but biological explanation is another.

Figs. 4 & 5: Ultimately, trees this high in elevation have usually shown positive climate associations with growing season temperature, as confirmed here.

Discussion: Again, do we not get to see a time series plot of temperature at or near the site?  Is it truly warming there?

A good number of references cited.  Though, it was surprising to see what was cited for COFECHA [28].  Those authors [28] may have mentioned COFECHA, but the usual citation for COFECHA has been Holmes 1983, the actual author of COFECHA.
